# Untargeted metabolomics confirms the association between plasma branched chain amino acids and residual feed intake in beef heifers

Ezequiel Jorge-Smeding[1,2], Sergio Polakof[3], Muriel Bonnet[1], Stephanie Durand[4], Delphine Centeno[4], Mélanie Pétéra[4], Sébastien Taussat[5,6], Gonzalo Cantalapiedra-Hijar[1]*

**1** INRAE, VetAgro Sup, UMR Herbivores, Université Clermont Auvergne, Saint-Genès-Champanelle, France, **2** Facultad de Agronomía, Departamento de Producción Animal y Pasturas, Universidad de la República, Montevideo, Uruguay, **3** INRAE, Unité de Nutrition Humaine (UNH), Université Clermont Auvergne, Clermont-Ferrand, France, **4** INRAE, UNH, Plateforme d'Exploration du Métabolisme, MetaboHUB Clermont, Université Clermont Auvergne, Clermont-Ferrand, France, **5** INRAE, AgroParisTech, GABI, Université Paris-Saclay, Jouy-en-Josas, France, **6** Eliance, Paris, France

* gonzalo.cantalapiedra@inrae.fr

**Data Availability Statement:** All relevant data are available within the Supporting Information.

## Abstract

This study explored plasma biomarkers and metabolic pathways underlying feed efficiency measured as residual feed intake (**RFI**) in Charolais heifers. A total of 48 RFI extreme individuals (High-RFI, n = 24; Low-RFI, n = 24) were selected from a population of 142 heifers for classical plasma metabolite and hormone quantification and plasma metabolomic profiling through untargeted LC-MS. Most efficient heifers (Low-RFI) had greater (P = 0.03) plasma concentrations of IGF-1 and tended to have (P = 0.06) a lower back fat depth compared to least efficient heifers. However, no changes were noted (P ≥ 0.10) for plasma concentrations of glucose, insulin, non-esterified fatty acids, β-hydroxybutyrate and urea. The plasma metabolomic dataset comprised 3,457 ions with none significantly differing between RFI classes after false discovery rate correction (FDR > 0.10). Among the 101 ions having a raw P < 0.05 for the RFI effect, 13 were putatively annotated by using internal databases and 6 compounds were further confirmed with standards. Metabolic pathway analysis from these 6 confirmed compounds revealed that the branched chain amino acid metabolism was significantly (FDR < 0.05) impacted by the RFI classes. Our results confirmed for the first time in beef heifers previous findings obtained in male beef cattle and pointing to changes in branched-chain amino acids metabolism along with that of body composition as biological mechanisms related to RFI. Further studies are warranted to ascertain whether there is a cause-and-effect relationship between these mechanisms and RFI.

## Introduction

In the current context of increasing food demand for a growing human population and feed to food competition, the efficient use of natural resources is a crucial challenge for beef

**Funding:** Laboratory analysis of this study were all funded by APISGENE. Ezequiel Jorge-Smeding received funds from Agencia Nacional de Investigacion e Innovacion (Uruguay) through the graduate scholarship (POS_NAC_2017_1_141119) and from the French government (Ambassade de France en Uruguay, Campus France) through the international mobility scholarship (928831G). The funders had no role in study design, data collection and analysis, decision to publish, or preparation of the manuscript.

**Competing interests:** None

production systems. Genetic selection for improved feed efficiency (defined as the ability of producing the same with the least feed resources) is a promising strategy to enhance beef cattle profitability given that the high inter-individual variability [1] and moderate heritability [2] of this animal trait. Among the different metrics that exists for measuring individual feed efficiency, residual feed intake (**RFI**) has gained notoriety because it does not entail a concomitant increase in adult body weight or mature body size compared to other metrics [3]. However, RFI determination requires expensive and long-lasting tests of dry matter intake which may cause additional concerns in such recordings in grazing conditions related to calf-cow systems [4]. Thus, in the light of new technology leading to high-throughput metabolic phenotyping, the exploration of biological circulating markers of RFI through metabolomics has emerged as a potential avenue to develop easier and cheaper strategies for genetic selection of superior animals, while revealing the physiological mechanisms underlying feed efficiency [5, 6].

Among the candidate RFI biomarkers identified in the last years through these metabolomics-based approaches, the branched-chain amino acids (**BCAA**) leucine, valine and isoleucine are being consistently reported [7–10]. Increased blood circulating concentrations of BCAA appear to be associated with lower feed efficiency (higher RFI values) possibly related with impaired insulin action and enhanced adiposity in less efficient animals [7, 8]; however, these results have been mostly obtained for male populations. Yet, metabolic differences across sex regarding the growing physiology [11] (*e.g.*: dysmorphic patterns in growth hormone and IGF-1 secretion) as well as the contrasted diet effect on growth performances between heifers and young bulls [12] may preclude any extrapolation from males to females. Indeed, the slower growth rates (and so the greater contribution of maintenance to total requirements), the earlier physiological maturity and the different body gain composition could affect feed efficiency differently between males and females [13]. In fact, McKenna et al. [14, 15] demonstrated that hepatic, skeletal muscle and adipose tissue differential gene expression between RFI classes is sex-dependent. In this sense, accurate RFI biomarkers for females could help to identify which heifers should be retained as replacement animals and which ones should be destined for fattening. Improving feed efficiency of females would impact the profitability of the whole beef cattle industry due to the significant contribution of females to total feeding costs [16]. Therefore, this work aimed at both exploring circulating candidate biomarkers of RFI and to unravel key metabolic pathways correlated with the observed differences in feed efficiency of growing beef heifers fed high-forage diets. To the best of our knowledge, our study is the largest experimental setup for searching RFI biomarkers in beef heifers through an untargeted metabolomic approach.

## Materials and methods

### Animals, diets and experimental design

The experiment was carried out in 4 periods (one cohort each) so-called: autumn 2013, winter 2013, autumn 2015 and winter 2015 at the experimental INRAE farm of 'Galle' (Bourges, Centre Region, France; https://doi.org/10.15454/1.5483259352597417E12). Animals were managed according to the INRAE ethics policy in agreement with the guidelines for animal research of the French Ministry of Agriculture. The approval number for ethical evaluation was APAFIS#14764–2018030610486896 v4.

One hundred and forty-two Charolais heifers, progeny of purebred Charolais cows inseminated with a set of 53 purebred Charolais bulls, were tested for RFI. Heifers were weaned at 7–8 months of age and kept unbred until the end of the RFI test. After a 4-week adaptation period, heifers entered in a 12-week RFI test (676 ± 16 days of age and 496 ± 50 kg of live weight).

Animals were always kept indoors in free stalls (three pens with a maximum of 12 heifers per pen) covered with wood shavings. Diet was offered *ad libitum* as a total-mixed ration (**TMR**) of grass silage and concentrate (95:5, respectively, in dry matter basis). Grass silage consisted of Fescue (*Festuca arundinaceae*), and concentrate was made from 29% dehydrated alfalfa (*Medicago sativa*) hay, 23% dehydrated beet pulp, 27% wheat middling and 12% cereal bran, plus minerals. Water was available ad libitum. Dry matter content and chemical composition of diet are shown in Table 1.

The feed was distributed once per day at 0800 am. Heifers were then blocked for 2 hours after meal distribution. Offered roughages and concentrates were weighed daily. Feed refusals were removed and weighed three times weekly (Monday, Wednesday and Friday). Dry matter (**DM**) contents of offered roughages and feed refusals were measured in a 103˚C ventilated oven during 24h.

Individual dry matter intake (**DMI**) was measured three times per week using individual troughs and electronically detected gates (American Calan Inc., Northwood, NH, USA). Fed body weight (**BW**) was measured twice at the start and at the end of the testing period, and BW was recorded fortnightly throughout the experiment. A regression of BW on test day was performed for each animal using a Proc REG of the SAS 15.1 statistical package (SAS Institute Inc., Cary, NC, USA). The start and mid-test BW were predicted when test day was set to zero and to half of the test duration, respectively. The regression slope was used as a measure of average daily gain (**ADG**). At the end of the test, fat thickness was ultrasonically measured on the back over three points: *gluteus medius* (**GM**) muscle at the 4th lumbar vertebra, *gluteus superficialis* (**GS**) muscle between the *Tuber coxae* and *Tuber ischiadicum* and, the *latissimus dorsi* (**LD**) muscle between the 12th and the 13th rib (ALOKA PROSOUND 2, DM Imaging, Montanay, France). These measurements included both the skin and the fat thicknesses.

The RFI was calculated as the residual from a multiple regression of DMI on metabolic body weight (MBW) and ADG in a model that included the pen within each period as a fixed effect: $DMI = Period \times Pen + \beta_1 MBW + \beta_2 ADG + RFI$. The variance of DMI within-pen and period explained by the performance variables included in the model was moderate ($r^2 = 0.30$) and all the terms were significant ($P < 0.05$) in the final model.

On the last day of the RFI test in the morning before meal distribution, blood was sampled from the coccygeal vein with 9 mL heparinzed vacutainer® tubes (BD vacutainer, Plymouth, UK) and immediately centrifuged ($2500 \times g$ for 15 minutes at 4˚C) for plasma isolation. Samples were stored at -80˚C until analysis. Based on RFI classification, the 12 most extreme heifers (High-RFI, n = 6; Low-RFI, n = 6) per period (autumn 2013, winter 2013, autumn 2015

**Table 1. Chemical composition and energy content of the experimental feed (mean ± SD).**

|  | Grass silage | Concentrate |
|---|---|---|
| Inclusion rate (%) | 95.3 ± 0.9 | 4.7 ± 1.0 |
| Dry matter (%) | 24.0 ± 2.9 | 88.9 |
| Crude protein (g/kg DM) | 111 ± 6 | 139 |
| Ash (g/kg DM) | 97.2 ± 6 | 87.1 |
| Neutral detergent fiber (g/kg DM) | 548 ± 22 | 427 |
| Cellulose (g/kg DM) | 305 ± 18 | 172 |
| Starch (g/kg DM) | — | 162 |
| NEg[1] (MJ/kg DM) | 6.04 ± 0.10 | 5.96 |

[1]Net energy for growth and maintenance according to Prevalim® module of the Inration v5® software (INRA, 2018)

and winter 2015) were selected adding a total of 48 (High-RFI, n = 24; Low-RFI, n = 24) animals used for exploring the plasma metabolome through LC-MS.

## Quantification of classical metabolic parameters in blood plasma

Blood plasma samples were subjected to classical metabolites and hormones quantification as previously reported [8]. In brief, the concentrations of glucose (glucose oxidase method), urea (gluta- mate dehydrogenase method), non-esterified fatty acids (**NEFA**, acyl-CoA synthase method) and BHB (D-$\beta$-hydroxbutyrate- dehydrogenase method) were determined using commercial kits (Thermo Scientific References #981379, #984325 and #981818 for glucose, BHB and urea, respectively and Sobioda Reference #W1W434-91795 for NEFA) and an autosampler spectrophotometer (Arena 20XT, Thermo Fisher Scientific, Cergy Pontoise, France). Plasma insulin (mean intra-assay coefficients of variation were 6.9% for 5.88 $\mu$UI/mL and 1.8% for 36.7 $\mu$UI/mL; Porcine Insulin RIA, MI-PI-12K, Merck KGaA, Darmstad, Germany) and IGF-1 (mean intra- and inter-assay coefficients of variation were 8.8% and 11.5%, respectively for 60.5 ng/mL; IGF-I RIA-CT, Ref IGF-R22, DiaSorin, Saluggia, Italy) were determined using radioimmunoassay.

## Metabolomic profiling of blood plasma and metabolomic data preparation

To explore blood plasma metabolites as possible candidate RFI biomarkers, the metabolomic profile was determined in the 48 extreme RFI selected heifers using a non-targeted approach based on LC-MS method. In brief, 100 uL of plasma samples were thawed at room temperature and mixed with 200 u$\mu$L of ice-cold methanol, vortexed and stored at -20˚C during 30 min. The mixture was then centrifuged for 10 min (15,000 rpm, 4˚C). 150 u$\mu$L of the supernatant were removed and dried off using a Genevac concentrator (EZ-2.3). Then, the samples were suspended in 500 u$\mu$L $H_2O$: acetonitrile (50:50) (v/v) and 0.1% formic acid. To ensure performance of analytical instrumentation and that the data were of comparable high quality through the analytical serie, quality control samples (**QC**) were inserted every 10 plasma samples. Each pooled QC was obtained by mixing 10 $\mu$L of each extracted plasma sample. These QC samples were processed as above. Metabolic profiles were acquired on a 26-min elution gradient on an U300 UHPLC system (Thermo Scientific) coupled to a high resolution QTOF (Bruker Impact HD2) operating in positive electrospray ionization mode, with a scan range from 50 to 1000 m/z. The capillary was set to 2.5 kV, the nebulizer was operated at 40 PSI, and the dry gas was set to 5 L/min at a temperature of 200˚C. Separation was performed on a 150 × 2.1 mm i.d. reverse-phase column (Acquity HSS T3, Waters) and using a water/acetonitrile (both containing 0.1% formic acid) gradient at a flow rate of 0.4 mL/min. The gradient was started at 100% water, held for 2 min, and decreased to 0% in 13 min; 0% water was maintained for 7 min before the gradient was returned to initial conditions and maintained for 4 min for re-equilibrating the column prior the next injection. The column temperature was set at 30˚C. Samples were kept at 4˚C during analyses.

Data were analysed using the open-source web-based interface Galaxy instance Workflow4-Metabolomics (https://workflow4metabolomics.org) running under R3.5.2 [17]. Pre-processing procedures were also applied to remove noise and unwanted variation including filtration and normalization. A data matrix containing mass and retention time with associated signal intensities for all remaining peaks was generated. After removing noise and filtering test, a total of 3,457 ions were finally extracted from the initial dataset comprising 10,585 ions. The dataset comprising the 3,457 ions extracted were further subjected to statistical analysis and subsequent metabolite annotation.

## Statistical analyses

As previously described [8], prior to statistical analyses all data, except RFI values, were adjusted for the period effect by using a linear model considering the cohort as the fixed effect and using R software v. 4.1.3. (//www.R-project.org/). After that, performance data and classical metabolites (glucose, NEFA, BHBA, urea) and hormones (insulin and IGF-1) were analysed by t-test, considering the RFI class as the only fixed effect. The significance threshold was set at $P \leq 0.05$, and tendency was declared at $0.05 < P \leq 0.10$.

Metabolomic data, after adjustment by the period effect, was analysed using the MetaboAnalyst v5 (https://www.metaboanalyst.ca/). The data was log-transformed prior to any analysis and subjected to multivariate and univariate analyses. Multivariate analysis included principal component analysis (**PCA**) and partial least square discriminant analysis (**PLSDA**). The PLSDA were performed on the basis of the entire metabolomic dataset (3,457 ions) or considering only those ions identified as metabolites (confirmed by standards; see further section on metabolite annotation) and differing between RFI class. The PLS-DA models were assessed according to its goodness of fitness ($\mathbf{R^2}$) and prediction quality index ($\mathbf{Q^2}$) where values close to 1.0 are preferred. Within each PLS-DA model, the metabolites' contribution to the model was assessed through the variable importance in projection (**VIP**) scores (the greater value, the highest contribution). Data was also submitted to univariate analysis through t-test by only considering the RFI class effect. The obtained raw-P values were adjusted by the Benjamini-Hochberg false discovery rate (**FDR**). Significance threshold was set at $FDR \leq 0.05$, and tendency was declared at $0.05 < FDR \leq 0.10$.

Finally, in order to get a deeper insight on possible biochemical mechanisms, metabolic pathways were explored through quantitative enrichment analysis based the *Kyoto Encyclopedia of Genes and Genomes* (KEGG) by using the *Globaltest* algorithm. In brief, Globaltest calculates the association between the metabolite sets and the phenotype without referring to a background [18]. Significant enrichment of metabolic pathway was set at $FDR \leq 0.05$, and only metabolic pathways with at least two metabolites measured in the current data set were further considered for discussion purposes.

## Metabolite annotation

The ions' annotation was carried out in two steps: first, a selection of ions based on the RFI effect (raw $P < 0.05$; n = 101) were subjected to putative annotation through an in-house database containing more than 1,000 metabolites analyzed under the same conditions and based on both retention time and m/z ratio obtained from the time-of-flight-MS. As a result, 13 out of 101 selected ions, were annotated by this approach and further subjected to identity confirmation (annotation class 1, according to Sumner et al. [19]) through fragmentation experiments by using compounds' standards. At the end, 8 out of 12 RFI discriminant ions were confirmed as metabolites (annotation class 1, according to Sumner et al. [19]). The 8 annotated ions corresponded to 6 metabolites (because two metabolites were identified through two different ion fragments).

# Results

## Animal performance and classical metabolic parameters

As expected, dry matter intake was lower (-32%, $P < 0.01$) for Low- vs High-RFI heifers whereas no significant differences were observed for BW or ADG (Table 2). Consequently, FCE was higher (+47%, $P < 0.01$) for Low- vs High-RFI group. The GM back fat thickness tended to be greater for High- than Low-RFI heifers, while the greater mean values for GS and LD anatomical locations could not be considered to be significant (Table 2).

**Table 2. Animal performance and classical plasma blood metabolites across Low vs. High residual feed intake (RFI) heifers.** Data is presented as least square means and standard error of the mean (SEM).

|  | *High-RFI* | *Low-RFI* | *SEM* | *P-value* |
|---|---|---|---|---|
| Dry matter intake, kg/d | 10.5 | 7.2 | 0.24 | < 0.01 |
| Residual feed intake, kg/ d | 1.59 | -1.77 | 0.221 | < 0.01 |
| Body weight, kg | 531 | 531 | 14.0 | 0.99 |
| Average daily gain, kg/d | 0.93 | 0.97 | 0.041 | 0.38 |
| Feed conversion efficiency, g/kg DMI | 89 | 131 | 4.8 | < 0.01 |
| Ultrasound back fat thickness[1], mm |  |  |  |  |
| GM | 9.1 | 8.5 | 0.41 | 0.06 |
| GS | 7.9 | 7.5 | 0.34 | 0.14 |
| LD | 7.1 | 6.9 | 0.25 | 0.42 |
| Plasma metabolites, mM |  |  |  |  |
| Urea | 2.50 | 2.33 | 0.017 | 0.58 |
| NEFA[2] | 0.37 | 0.34 | 0.026 | 0.19 |
| BHB[3] | 0.39 | 0.36 | 0.020 | 0.10 |
| Glucose | 4.40 | 4.35 | 0.088 | 0.56 |
| Plasma hormones |  |  |  |  |
| Insulin, µU/mL | 15.0 | 15.6 | 2.12 | 0.78 |
| IGF-1[4], ng/mL | 207 | 237 | 13.3 | 0.03 |

[1]GM: *Gluteus medium*; GS: *Gluteus superficialis*; LD: *Latissimus dorsi*

[2]NEFA: non-esterified fatty acids

[3]BHB: β-hydroxybutyrate

[4]IGF-1: insulin-like growth factor 1

Plasma concentrations of urea, NEFA, glucose, BHB and insulin did not differ between RFI classes ($P \geq 0.10$). In contrast, plasma concentrations of IGF-1 was higher in Low-RFI heifers than their High-RFI counterparts (Table 2).

## Metabolomic profiling: Multivariate and univariate analyses

A total of 3,457 ions were measured and retained in the dataset after filtering and cleaning procedures of raw data. On the basis of this entire dataset (3,457 ions), the main variabilities highlighted by the PCA did not match the RFI class (Fig 1). In addition, no valid predictive model was obtained by PLS-DA analysis ($Q^2 < 0$ for first component; S1 Fig) neither using the total metabolomic dataset (3,457 ions) nor the annotated and confirmed metabolites (annotation class 1).

According to t-test analysis, no ion out of the 3,457 was found to differ among RFI classes after Benjamini-Hochberg correction (FDR > 0.10 in all cases). However, 101 ions were observed to have a raw-$P \leq 0.05$. Eight out these 101 ions were annotated as compounds (class 1 annotation), corresponding to 6 metabolites (two metabolites were identified by two ion fragments), including leucine, valine, lysine, cytidine, ornithine and methylimidazolacetic acid (Table 3).

To note, signal intensity of leucine, valine, lysine and ornithine appeared to have higher mean values in High-RFI heifers in this experiment while cytidine and methylimidazolacetic acid gave higher mean values in Low-RFI animals (Table 3, Fig 2).

## Metabolic pathways analysis

Metabolic pathway analysis performed on the basis of identified compounds (class 1 annotation) showed several pathways differing (FDR $\leq$ 0.05) between RFI groups. Most of them (9/

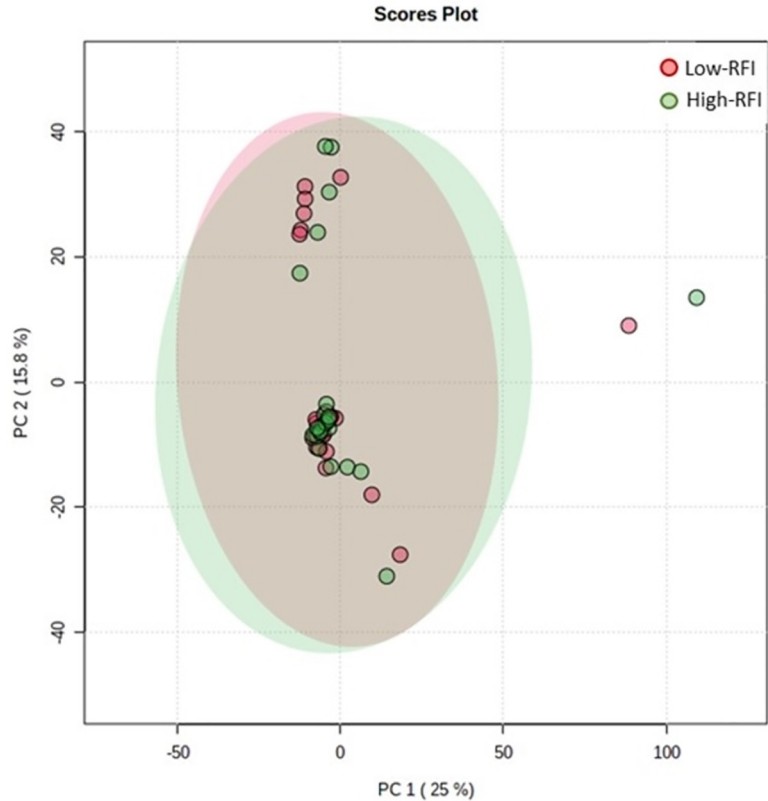

**Fig 1. Principal component analysis plot score according to residual feed intake class (Low vs. High RFI) of Charolais heifers.**

10) belonged to AA metabolism such as the metabolism of BCAA or histidine. However, only the BCAA metabolism (degradation and synthesis), and aminoacyl-tRNA biosynthesis were identified on the basis of at least two compounds (Table 4) being both of these pathways related with increased plasma concentrations of leucine and valine for High- vs. Low-RFI heifers.

**Table 3. Metabolites identified through ion annotation and further confirmed by standards which were affected (raw-P < 0.05) by residual feed intake class (Low vs High) in beef heifers.**

| Ion | Metabolite[1] | m/z[2] | rt (min)[3] | P-value |
|---|---|---|---|---|
| M112.05052T1.21 | Cytidine | 112.0505 | 1.21 | 0.01 |
| M132.10185T2.54 | Leucine$_1$ | 132.1018 | 2.54 | 0.01 |
| M86.09637T2.55 | Leucine$_2$ | 86.0963 | 2.55 | 0.01 |
| M147.1128T0.78 | Lysine | 147.1128 | 0.78 | 0.02 |
| M141.06598T1.09 | Methylimidazolacetic acid | 141.0660 | 1.09 | 0.02 |
| M115.08654T0.78 | Ornithine$_1$ | 115.0865 | 0.78 | 0.03 |
| M133.0972T0.78 | Ornithine$_2$ | 133.0972 | 0.78 | 0.01 |
| M72.08057T1.21 | Valine | 72.0806 | 1.21 | 0.03 |

[1] numbers in subscript are depicted when one metabolite was identified through two ion fragments

[2] mass:charge ratio

[3] retention time

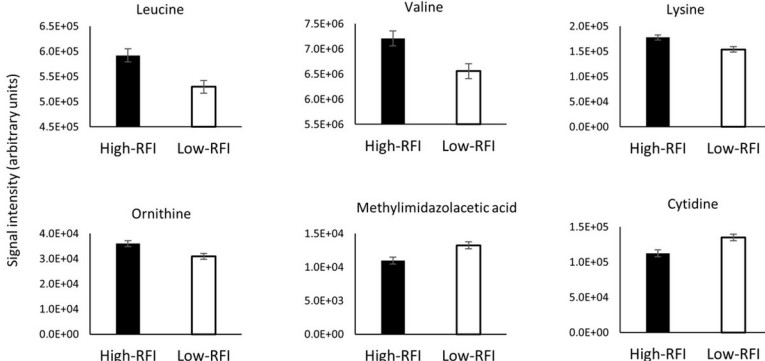

**Fig 2. Metabolites identified through ion annotation and further confirmed by standards which were affected (raw-P < 0.05) by residual feed intake class (Low vs High) in beef heifers.** High and Low-RFI classes are depicted by black and empty bars, respectively. For leucine and ornithine, only the ion with the highest peak intensity was plotted.

## Discussion

Recently published studies regarding RFI biomarkers are consistently identifying the blood circulating BCAA levels to be positively associated with RFI in entire or castrated male beef cattle [7, 8, 10]. Similar results were reported by Li et al. [9] on a mixed population of male and female beef cattle; however, to the best of our knowledge, accurate RFI plasma biomarkers for beef heifers are still lacking. Our metabolic pathway analysis, conducted from a set of annotated and confirmed compounds, found a significant relationship (FDR < 0.05) between BCAA metabolism and RFI in growing heifers as previously reported in male beef cattle [20]. As further discussed, our results suggested that increased BCAA plasma levels in less efficient animals could be associated with increased lipid synthesis and fat accretion.

Metabolomic data comparing RFI classes in beef males belonging to both British and continental breeds highlighted plasma BCAA as a biomarker discriminating Low- vs. High-RFI animals with increased circulating levels in less efficient animals [7–10]. In this sense, our results confirm similar trends when comparing Low- vs. High-RFI Charolais heifers. Despite similar results were reported by Li et al. [9] when using a female and male mixed population, the sex effect was not tested, being our study the first one to specifically report BCAA differences among RFI classes in beef heifers. Despite the biological links between BCAA and RFI is far from being understood in ruminants, results from the above-mentioned studies suggest that it might include increased adiposity accretion and lower insulin sensitivity through the activation of the mechanistic target of rapamycin (**mTOR**). In this regard, the literature on ruminants and non-ruminants have established that the BCAA can act as signalling molecules sensing the nutritional state (i.e. balance between nutrient absorption and animal

**Table 4. Metabolic pathways identified to be differentially altered among contrasting residual feed intake (RFI) Charolais heifers.**

| Metabolic pathway | Total comp[1] | Hits[2] | *Raw P-value*[3] | *FDR*[3] |
|---|---|---|---|---|
| Valine, leucine and isoleucine degradation | 40 | 2 | < 0.01 | 0.03 |
| Valine, leucine and isoleucine biosynthesis | 8 | 2 | < 0.01 | 0.03 |
| Aminoacyl-tRNA biosynthesis | 48 | 3 | 0.01 | 0.03 |

[1]Total metabolites theoretically considered by the KEGG database for the current metabolic pathway

[2]Metabolites effectively quantified in the current study and belonging to the identified pathway

[3]raw-P and FDR values obtained with the Global Test

requirements) and likely participating in the initiation of some metabolic process including insulin signalling [21], stimulation of fatty acid synthesis [22] and ingestive behaviour regulation [23] through the activation of mTOR in the peripheral tissues [24]. In the current study, the trend for a higher backfat thickness in less efficient RFI heifers confirms previous findings of increased plasma BCAA in less efficient bulls in association with higher adiposity [8]. As reviewed by Newgard [25], increased circulating levels of BCAA likely activating the mTOR pathway are associated with the incidence of higher obesity and type 2 diabetes in humans as widely evidenced during the last decades in several studies, and it has been established that high BCAA can indirectly undermine insulin sensitivity through the chronic activation of the mTOR pathway, inducing an over phosphorylation of the insulin receptor substrate [26, 27]. Regarding bovines, it has been demonstrated that mTOR is upregulated in tissues from inefficient RFI beef cattle [28]. In the same line, it was reported that higher plasma BCAA concentrations were associated with higher body condition score in dairy cows during the dry period [29], while some authors found that BCAA abundance could regulate lipogenesis through mTOR and protein kinase B activation [30]. In the current study, the higher DMI in less efficient heifers leading to increased nutrient availability could have somehow led to a chronic activation of the mTOR pathway and thus to a lower insulin sensitivity [25, 26] and higher BCAA catabolism [30]. In fact, it has been demonstrated that during adipogenesis the BCAA catabolism is up regulated in murine adipocytes [31]. We hypothesise that increased BCAA levels inducing enhanced BCAA catabolism and leading to higher abundance of branched-chain keto acids could have stimulated, at least partially, the higher adiposity of High-RFI animals. In agreement with our current results and other metabolomic-based experiments, a meta-analysis based on ten genome-wide association studies in beef cattle identified the BCAA degradation as the only metabolic pathway related to RFI [20]. Similarly, Khansefid et al. [32] reported the BCAA degradation as a pathway differentially expressed in Low- vs. High-RFI Angus bulls (liver and muscle tissue), and Holstein cows (liver and white blood cells).

It is interesting to note that in the current study the heifers were fed a high-forage low-energy diet while those studies reporting positive relationships between RFI and BCAA in males have been performed almost exclusively with energy-dense diets likely leading to insulin sensitivity differences between RFI groups. In our previous study in Charolais young bulls, the higher plasma BCAA concentrations and adiposity in High-RFI animals were associated with higher plasma insulin in a diet-dependent manner as it was only observed in a high-starch diet but not in a high-fiber diet [8]. Our data together with those from the literature point at plasma BCAA concentrations to be directly or indirectly associated with differences in lipid synthesis and fat accretion both in females (present study) and males beef cattle. However, it is worth to keep in mind that the cause-effect relationship between BCAA circulating levels and adiposity are not yet fully understood. Indeed, the crosstalk between BCAA and adipose tissue metabolism appear to be interconnected in a vicious circle where higher feed intake and obesity stimulate BCAA concentrations raise and catabolism, while BCAA catabolites accumulation may impair insulin function (reviewed by Nie et al. [33]).

The differences in plasma IGF-1 and methylimidazole-acetic acid could be also related with differences in body composition between RFI groups. The higher IGF-1 observed in the Low-RFI heifers could be related with their leaner body mass since this hormone mediates the anabolic action of the growth hormone and insulin in several tissues, including the skeletal muscle, playing a central role in nutrient metabolism [34]. In feed efficiency studies of beef cattle, IGF-1 has been suggested to be associated with the between-animal variation in feed efficiency in some reviewed studies [35]. In this sense, our results agree with previous studies reporting increased plasma concentrations of IGF-1 and higher muscle gene expression of IGF-1 receptor for Low- vs. High-RFI heifers [36, 37]. Indeed, plasma IGF-1 was proposed as a biomarker

of RFI because its stimulatory effect on protein synthesis [38] and so possibly linked with the leaner body mass usually observed in Low-RFI animals [35]. However, the diet may also have an impact on the relationship between RFI and IGF-1, as reviewed by Cantalapiedra-Hijar et al. [35] since negative associations between IGF-1 and RFI have been reported in studies using low-concentrate and/or low-energy diets while in high-concentrate diets this relationship has been shown to be inconsistent. Additionally, despite it has been widely reported that Low-RFI animals have usually leaner body composition [35], some studies have suggested that differences in adiposity between RFI classes might change according to the diet, with no significant differences observed in fattening bulls fed high-fibre total mixed diets [8] or even higher adiposity in Low- than High-RFI beef heifers when tested under grazing conditions [39].

The increased plasma levels of methylimidazole-acetic acid found in Low-RFI vs High-RFI heifers seemed to agree with their leaner body composition. Methylimidazole-acetic acid is the main catabolic product of histamine normally excreted in urine [40]. In humans, greater concentrations of methylimidazole-acetic acid in males have been associated with its larger body size compared to women, but gender differences disappear when corrected for creatinine [41]. In the current experiment RFI groups did not differ in their BW; thus, we suggest that higher plasma concentrations of methylimidazole-acetic acid could be indirectly translating increased lean body mass. Indeed, in humans it has been reported that methylimidazole acetic acid is associated with body weight relative to body mass index, with greater concentrations of this metabolite associated with greater body weight but lower body fat [40]. Additionally, the differences in this metabolite between RFI groups could be also indicative of a regulatory role on the control of the voluntary intake. Yoshimoto et al. [42] demonstrated that increased histamine concentrations downregulate the appetite in mice and it has been reported that the QTL regions of histamine receptor in the brain are associated with RFI class both in Nellore cattle and broiler chickens possibly addressing a control centre of the ingestive behaviour [43, 44]. Thus, we could conjecture that if plasma concentrations of methylimidazole-acetic acid are reflecting increased histamine catabolism it could also suggest higher circulating histamine and then more important amounts of this amino acid reaching the brain and signalling for a higher satiety in Low-RFI heifers.

Among the suggested mechanisms underlying RFI, urea synthesis has been discussed by different authors [45, 46]; however, the evidence is scarce until now. When looking at ornithine, a key metabolite in the urea cycle, in the current study the reported mean of plasma levels was lower for Low-RFI animals, possibly indicating decreased urea cycle activity. Positive associations between urea plasma concentrations and RFI have been previously reported in steers [47, 48] and young bulls [49, 50], however in the current study no changes in plasma urea concentration were observed across RFI classes. Interestingly, Gonano et al. [51] when comparing blood profiles of RFI groups in beef heifers with different physiological stages found lower plasma urea for Low- vs. High-RFI only in pregnant heifers. In the light of these results, it is possible that the lack of significant differences in our study could be due to a modulatory effect of the physiological stage interacting with RFI phenotype on circulating urea levels, with more evident differences when nitrogen requirements are increased (such as during pregnancy). Notwithstanding before, our results agree with those recently reported by Hashemiranjbar [52] for dairy primiparous cows who also observed decreased ornithine and lysine plasma concentrations in Low- vs. High-RFI dairy heifers at 150 days in milk, and a tendency for lower plasma concentrations of urea and lysine for the same animals later on lactation at 240 days in milk. In fact, lysine and valine which were lower for Low- than High-RFI heifers can enter into the urea cycle through their transamination to glutamate which can fuel the urea cycle through its enchained transformation to N-acetylglutamate and carbamoyl-phosphate [53]. As a whole, lower plasma ornithine, valine and lysine in the Low-RFI heifers in the

current study could be consistent with a decreased urea synthesis. Moreover, reduced ureagenesis might reflect increased nitrogen use efficiency in the most efficient animals [35, 54] as recently demonstrated in young bulls [49, 50]. In fact, we have recently observed in another RFI experiment lower plasma urea concentration in Low-RFI young bulls suggesting—together with metabolomic data—improved N use efficiency in low RFI animals [8] which was further demonstrated through the use of natural $^{15}$N abundance in plasma as a biomarker of N use efficiency in ruminants [50].

## Limitations

We acknowledge that the lack of FDR significant differences for individual metabolites according to t-test is a limitation of the results discussed above, because of the increased risk of false positives. However, some metabolites showing raw-P values < 0.05 were aligned with the FDR-significant pathway related to BCAA metabolism (biosynthesis and degradation) and agree with other reported studies in beef cattle [19]. Thanks to the annotation and confirmation of a set of metabolites with raw P < 0.05 we succeeded to identify significant metabolic pathways that otherwise they would not have been identified with the stringent cut-off imposed by FDR corrections. In this regard, our untargeted metabolomic study was carried out in an explorative way to highlight candidate biomarkers and metabolic pathways rather than testing specific hypothesis such as, for instance, the confirmation of known biomarkers. Indeed, false positives can be always corrected by further investigation, whereas an experiment with a false negative result might never be repeated, and possible true treatment effects missed. As suggested by Lieberman et al. [55], there is traditionally too great emphasis on avoiding false positives, and that greater attention should be given to avoiding false negatives. As previously discussed, those metabolites with raw-P < 0.05 agreed with previous reported data studying the metabolic differences underlying RFI phenotype in beef cattle and were consistent with FDR-significant metabolic pathways, suggesting that true differences in its concentrations existed in the current study.

Another limitation of our study, common to classical experiments aiming at identifying mechanisms and biomarkers of RFI, is that its design does not allow to ascertain whether the identified metabolic pathways are really involved in some improvement of nutrient use efficiency or are only correlated to RFI because they co-vary with feed intake. Indeed, because BCAA plasma concentration are sensitive to nutrient supply and feed intake [56], we cannot conclude about a cause-and-effect relationship between BCAA metabolism pathway and RFI variations. This concern has been previously discussed in relation to biological determinants of RFI in the review by Cantalapiedra-Hijar et al. [35] and solutions to overcome this issue should involve feeding at the same feeding level (%BW) extreme RFI cattle from either genetically divergent lines or previously phenotypically ranked animals. Well-designed experiments are warranted in the future to dissociate a simple correlation between biomarkers and RFI from a cause-and-effect relationship.

## Conclusion

Despite the lack of individual metabolites differing (FDR < 0.05) between RFI groups, our metabolic pathway analysis, conducted from a set of annotated compounds, found a significant relationship (FDR < 0.05) between BCAA metabolism and RFI in growing heifers as previously reported in male beef cattle. Although the underlying mechanism is not completely understood, our results suggested that increased BCAA plasma levels in less efficient animals could be associated with increased lipid synthesis and fat accretion. In addition, other compounds possibly reflecting lean body mass such as IGF-1 were also decreased for High-RFI

heifers. Taken together, our results suggested that in beef heifers, the mechanisms underlying RFI may comprise the signalling role of BCAA and lipogenesis differences. Further studies are needed to mechanistically establish the link between BCAA and RFI, and to validate BCAA as predictive biomarkers of RFI in field conditions as support of genetic selection programs.

## Supporting information

**S1 Fig.** Partial least square discriminant analysis (PLS-DA) for residual feed intake (RFI) on the basis of: a) total metabolomic dataset (3,457 ions) or b) only confirmed metabolites (annotation class 1). Score plot, and cross validation values are presented in the top and bottom of each figure panel, respectively. Red and green dots depict Low- and High-RFI, respectively. (JPG)

**S1 File.**
(XLSX)

## Acknowledgments

We wish to thank to Marine Gauthier, Céline Chantelauze and Arnaud Delavaud (UMRH) and David Maupetit (INRAE-Bourges) for their great technical support. Metabolomic analysis were performed within the metaboHUB French infrastructure (ANR-INBS-0010).

## Author Contributions

**Conceptualization:** Sergio Polakof, Gonzalo Cantalapiedra-Hijar.

**Formal analysis:** Ezequiel Jorge-Smeding, Stephanie Durand, Delphine Centeno, Mélanie Pétéra, Sébastien Taussat.

**Funding acquisition:** Gonzalo Cantalapiedra-Hijar.

**Investigation:** Muriel Bonnet, Stephanie Durand, Delphine Centeno, Mélanie Pétéra, Sébastien Taussat, Gonzalo Cantalapiedra-Hijar.

**Supervision:** Gonzalo Cantalapiedra-Hijar.

**Writing – original draft:** Ezequiel Jorge-Smeding, Gonzalo Cantalapiedra-Hijar.

**Writing – review & editing:** Sergio Polakof, Muriel Bonnet, Stephanie Durand, Delphine Centeno, Mélanie Pétéra, Sébastien Taussat.

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
