## [Decision Letter · Decision Letter 0]

5 Aug 2022

PONE-D-22-15718Untargeted metabolomics confirms the association between plasma branched chain amino acids and residual feed intake in beef heifersPLOS ONE

Dear Dr. CANTALAPIEDRA-HIJAR,

Thank you for submitting your manuscript to PLOS ONE. After careful consideration, we feel that it has merit but does not fully meet PLOS ONE’s publication criteria as it currently stands. Therefore, we invite you to submit a revised version of the manuscript that addresses the points raised during the review process.

Some methodological and typographical problems were identified. The MS should be revised by eliminating these concerns. Please submit your revised manuscript by Sep 19 2022 11:59PM. If you will need more time than this to complete your revisions, please reply to this message or contact the journal office at plosone@plos.org. Please include the following items when submitting your revised manuscript:A rebuttal letter that responds to each point raised by the academic editor and reviewer(s). You should upload this letter as a separate file labeled 'Response to Reviewers'.A marked-up copy of your manuscript that highlights changes made to the original version. You should upload this as a separate file labeled 'Revised Manuscript with Track Changes'.An unmarked version of your revised paper without tracked changes. You should upload this as a separate file labeled 'Manuscript'.

We look forward to receiving your revised manuscript.

Kind regards,

Arda Yildirim, Ph.D.

Academic Editor

PLOS ONE

Journal Requirements:

“Laboratory analysis of this study were all funded by APISGENE. Ezequiel Jorge-Smeding received funds from Agencia Nacional de Investigacion e Innovacion (Uruguay) through the graduate scholarship (POS_NAC_2017_1_141119) and from the French government (Ambassade de France en Uruguay, Campus France) through the international mobility scholarship (928831G). “            

“None”

“We wish to thank APISGENE for their financial support of this project, which forms part of the larger national program BEEFALIM 2020. Thanks to Marine Gauthier, Céline Chantelauze and Arnaud Delavaud (UMRH) and David Maupetit (INRAE-Bourges) for their great technical support.”

“Laboratory analysis of this study were all funded by APISGENE. Ezequiel Jorge-Smeding received funds from Agencia Nacional de Investigacion e Innovacion (Uruguay) through the graduate scholarship (POS_NAC_2017_1_141119) and from the French government (Ambassade de France en Uruguay, Campus France) through the international mobility scholarship (928831G).“

6. Please upload a copy of Supporting Information S1 Fig which you refer to in your text on page 18.

Additional Editor Comments:

Dear Authors,

According to the feedback from the referees, some methodological and typographical problems have been identified in this MS. It needs to be revised to address the referees' suggestions and concerns. Thanks for hard work in advance.

Reviewers' comments:

Reviewer's Responses to Questions

**Comments to the Author**

1. Is the manuscript technically sound, and do the data support the conclusions?

Reviewer #1: No

Reviewer #2: Yes

Reviewer #3: Yes

2. Has the statistical analysis been performed appropriately and rigorously? 

Reviewer #1: No

Reviewer #2: Yes

Reviewer #3: Yes

3. Have the authors made all data underlying the findings in their manuscript fully available?

Reviewer #1: No

Reviewer #2: Yes

Reviewer #3: Yes

4. Is the manuscript presented in an intelligible fashion and written in standard English?

Reviewer #1: Yes

Reviewer #2: Yes

Reviewer #3: Yes

5. Review Comments to the Author

Reviewer #1: This manuscript describes a study to determine metabolite differences between low and high RFI heifers. This research will add to the scientific literature but I have reservations about the results. A false discovery rate was used to account for multiple treatment comparisons and no significant differences were found so the authors went back to the raw P-values to find metabolites to further evaluate. IF the FDR was necessary to begin with then why is it acceptable to go back and use raw P-values when the FDR does not result in any significant differences? This is not acceptable. However, the metabolites that were found to be different are consistent with previous literature.

As with a lot of RFI literature, there is a strong connection with DMI. The metabolites identified in this study are also involved in appetite regulation, but may also be involved in nutrient utilization. The objective of this study was to 1) identify biomarkers of RFI, which fits with the experimental approach, and 2) unravel key metabolic pathways explaining differences in feed efficiency, which is confounded by differences in DMI in this experimental approach. When low and high RFI animals are fed ad libitum, it is impossible to ascertain whether the differences between groups is due to differences in DMI, nutrient utilization efficiency, or some combination of both. The discussion in this paper is warranted, but the authors should be careful not to imply that the differences are due to appetite regulation or nutrient use efficiency because this cannot be determined. Further research on RFI really must separate the effects of appetite regulation and nutrient use efficiency between low and high RFI animals to answer the question of whether low RFI animals are really more metabolically efficient.

Specific comments:

L35 - there is a real methodological problem with reverting back to raw P-values after the FDR returned no significant results

L88 - what is unique about these bulls?

L90 - I am not intimately familiar with cattle genetics in France, but 1100-lb heifers at 9 months of age seems extreme.

L127 - why was the skin thickness included?

L128 - why not use RFI as the abbreviation for 'residual DMI)?

Table 2 - I assume this feed conversion efficiency is expressed as grams of gain per kg of DMI, but it needs to be specified.

L258 - Figure S1 is not part of the document and I could not find it on the editorial manager website

L377 - 'observed in Low-RFI' what - need a subject for the adjective

L399 - change 'indicting' to 'indicating'

Figure 1 PCA - 2 data points seem to be very much outliers - one from low rfi and one from high rfi. Were they evaluated as potential outliers? Why did they have such extreme values for PC 1?

Figure 2 - y-axis needs a title with units of measure

Reviewer #2: Pg 3, line 52 change arise to cause or arouse

Pg 4 where BW measurements shrunk weights?

Pg 4 Was sire tested in the statistical model and eliminated?

Pg 6 Why did you use porcine RIA instead of bovine RIA for insulin determinations?

Pg 9 define KEGG

Discussion: This paper confirms the fact that at a younger age, LRFI heifers are generally leaner. However, numerous studies show that aged LRFI cows tend to have greater body condition than due HRFI. This should probably be referenced somewhere in the discussion.

This paper adds value to the body of published research.

Reviewer #3: The manuscript has a large amount of useful information, mainly relationship between BCAA plasma levels and residual feed intake in growing cattle, focusing in heifers. The procedures are described in great detail, for that reason is worth of publication.

6. PLOS authors have the option to publish the peer review history of their article (what does this mean?). If published, this will include your full peer review and any attached files.

Reviewer #1: No

Reviewer #2: No

Reviewer #3: No

---

## [Author Response · Author response to Decision Letter 0]

18 Sep 2022

Response to reviewers

Reviewer #1: This manuscript describes a study to determine metabolite differences between low and high RFI heifers. This research will add to the scientific literature but I have reservations about the results. A false discovery rate was used to account for multiple treatment comparisons and no significant differences were found so the authors went back to the raw P-values to find metabolites to further evaluate. IF the FDR was necessary to begin with then why is it acceptable to go back and use raw P-values when the FDR does not result in any significant differences? This is not acceptable. However, the metabolites that were found to be different are consistent with previous literature.

AU: We agree with the reviewer and acknowledge that one limitation of our conclusions was the lack of FDR significant differences in the t-test for individual metabolites. However, this experiment was carried out to explore and identify candidate biomarkers rather than testing specific hypothesis on confirming known biomarkers. False positives can be always corrected by further investigation, whereas an experiment with a false negative result might never be repeated, and possible true treatment effects missed. Moreover, untargeted metabolomics approaches are usually selected in explorative studies with the aim to generate hypotheses for future targeted studies (Alexandra et al., 2016, J. Am. Soc. Mass Spectrometry, 10.1007/s13361-016-1469-y) such our case were almost no study has been published on beef female RFI biomarkers. In this regard, it has been suggested that there is traditionally too great emphasis on avoiding false positives, and that greater attention should be given to avoiding false negatives (Lieberman MD, Cunningham WA Soc Cogn Affect Neurosci. 2009 Dec; 4(4):423-8.). This is why we still consider to discuss the metabolites identified at the level of raw-P < 0.05. It is important to note that these FDR values are obtained on the basis of the 3457 detected ions which means very stringent cut-off in terms of raw-P values for a sample size of as low as 24 individuals per treatment. As the reviewer has highlighted, metabolites that were found to be different in our study on the basis of raw P-values are well aligned with previously reported data and when it comes to the metabolic pathway analysis, they seem to be related in common pathways that are indeed identified at FDR<0.05. 

 Following your criticism on this point we have decided to clearly state in the discussion and conclusion that: i) no significant individual metabolites were found to differ across RFI groups but that the analysis of a reduced set of annotated metabolites with raw-P <0.05 allowed us to find FDR significant pathways that otherwise would not have been identified, and ii) highlight the limitations, but also the consistency with other reports, of our interpretations. In this regard, a section about the limitation of our study has been included at the end of the discussion (L433-463) for addressing the issue about raw P value and the inability of our experimental design to ascertain whether the identified biomarkers/metabolic pathways underly RFI variations or are only correlated responses to DMI variation. 

As with a lot of RFI literature, there is a strong connection with DMI. The metabolites identified in this study are also involved in appetite regulation, but may also be involved in nutrient utilization. The objective of this study was to 1) identify biomarkers of RFI, which fits with the experimental approach, and 2) unravel key metabolic pathways explaining differences in feed efficiency, which is confounded by differences in DMI in this experimental approach. When low and high RFI animals are fed ad libitum, it is impossible to ascertain whether the differences between groups is due to differences in DMI, nutrient utilization efficiency, or some combination of both. The discussion in this paper is warranted, but the authors should be careful not to imply that the differences are due to appetite regulation or nutrient use efficiency because this cannot be determined. Further research on RFI really must separate the effects of appetite regulation and nutrient use efficiency between low and high RFI animals to answer the question of whether low RFI animals are really more metabolically efficient.

AU: We completely agree with this comment and thus we decided to change the second objective to “unravel key metabolic pathways correlated with the observed differences in feed efficiency” (L78-79). Our experimental design, and that from most (if not all) studies aiming at identifying candidate biomarkers of RFI, does not allow to ascertain whether plasma metabolites and identified metabolic pathways are associated with RFI only because they covary with DMI variation or because they are involved in changes in nutrient utilization. We now acknowledge in the discussion (in the new section about limitations of our study) that more studies are warranted to confirm a cause-and-effect relationship between the identified candidate biomarkers and RFI and that our experimental design does not allow to ascertain whether the identified candidate biomarkers for RFI are involved in a better nutrient use efficiency, changes in feeding level or both (L452-463). 

Specific comments:

L35 - there is a real methodological problem with reverting back to raw P-values after the FDR returned no significant results

AU: As previously discussed if we had not looked at those ions with raw P-values lower than 0.05, beyond the non-significant FDR results, we would not have annotated and thus identified metabolites that were already reported in the literature as candidate biomarkers of RFI in beef males. In the new submitted version, we have discussed the limitation of our study, acknowledged the inherent limits and changed the conclusions (L466-468) and the abstract (L35-37) for only speaking about significant (FDR<0.01) pathways. 

L88 - what is unique about these bulls?

AU: The word “unique” was removed. 

L90 - I am not intimately familiar with cattle genetics in France, but 1100-lb heifers at 9 months of age seems extreme.

AU: Please note that as stated in the original version, animals were around 22.5 months old (676 days) and their weight (496 kg) corresponded to a standard weight for Charolais heifers of that age. 

L127 - why was the skin thickness included?

AU: When the first RFI tests were performed in 2013, the available ultrasound equipment did not allow to isolate in a repeatable way the skin from the adipose tissue layer while both together gave repeatable thickness measurements. Later, with more sophisticated equipment, we succeeded to separate both and realized that skin thickness is little variable from one animal to another.

L128 - why not use RFI as the abbreviation for 'residual DMI)?

AU: Agree, this has been changed according to your suggestion (L134).

Table 2 - I assume this feed conversion efficiency is expressed as grams of gain per kg of DMI, but it needs to be specified.

AU: Corrected, thanks. 

L258 - Figure S1 is not part of the document and I could not find it on the editorial manager website

AU: Sorry, there must has been a problem during submission. We’ll make our best to ensure that this figure is available for reviewers in the new submission.

L377 - 'observed in Low-RFI' what - need a subject for the adjective

AU: Please note that this referred to methylimidazole-acetic acid: “The increased plasma levels of methylimidazole-acetic acid observed in Low-RFI…” For a better understanding we slightly changed the sentence to “The increased plasma levels of methylimidazole-acetic acid found in Low-RFI vs High-RFI heifers seemed to agree with their leaner body composition” (L388-389)

L399 - change 'indicting' to 'indicating'

AU: Thanks, corrected. 

Figure 1 PCA - 2 data points seem to be very much outliers - one from low rfi and one from high rfi. Were they evaluated as potential outliers? Why did they have such extreme values for PC 1?

AU: Effectively, these animals were evaluated as outliers but no clear evidence arose from heatmaps and univariate analysis. Actually, no evidence of uniqueness of these animals compared to the rest were obtained on the basis of phenotypic and field registers. 

Figure 2 - y-axis needs a title with units of measure

AU: Corrected

Reviewer #2: 

Pg 3, line 52 change arise to cause or arouse

AU: Thanks, corrected

Pg 4 where BW measurements shrunk weights?

AU: No, BW corresponded to fed BW and not shrunk BW. This is now clearly stated in the new version (L120-121). 

Pg 4 Was sire tested in the statistical model and eliminated?

AU: We are not sure about the purpose of such approach. If the effect of sire was considered in the RFI model we would have removed to some extent the genetic contribution to RFI variation. We aimed to identify candidate biomarkers of phenotypic RFI variation, the latter depending to some extent on genetics. Therefore, sire was not tested in the model and its effect was not consequently removed from residuals. 

Pg 6 Why did you use porcine RIA instead of bovine RIA for insulin determinations?

We used porcine RIA kit for insulin determinations as there is not currently RIA kit for bovine. Among those available, the porcine RIA kit has the highest affinity for ruminants (90% of specificity). 

Pg 9 define KEGG

AU: Done. 

Discussion: This paper confirms the fact that at a younger age, LRFI heifers are generally leaner. However, numerous studies show that aged LRFI cows tend to have greater body condition than due HRFI. This should probably be referenced somewhere in the discussion.

AU: Thank you for your comment. We have added some new elements in the discussion supporting that no differences or even higher adiposity in Low RFI beef cattle can be observed in particular feeding conditions. As far as one objective of RFI biomarkers discovery is to contribute to early selection of superior animals we think that it may be more relevant to discuss the possible effects of the diet rather than the age itself (L383-387). 

This paper adds value to the body of published research.

Reviewer #3: The manuscript has a large amount of useful information, mainly relationship between BCAA plasma levels and residual feed intake in growing cattle, focusing in heifers. The procedures are described in great detail, for that reason is worth of publication.

AU: Thank you for your positive feedback

---

## [Decision Letter · Decision Letter 1]

28 Oct 2022

Untargeted metabolomics confirms the association between plasma branched chain amino acids and residual feed intake in beef heifers

PONE-D-22-15718R1

Dear Dr. CANTALAPIEDRA-HIJAR,

We’re pleased to inform you that your manuscript has been judged scientifically suitable for publication and will be formally accepted for publication once it meets all outstanding technical requirements.

Kind regards,

Arda Yildirim, Ph.D.

Academic Editor

PLOS ONE

Additional Editor Comments (optional):

Thanks to the Authors that have adequately addressed the comments previously done by the different reviewers.

Reviewers' comments:

Reviewer's Responses to Questions

**Comments to the Author**

1. If the authors have adequately addressed your comments raised in a previous round of review and you feel that this manuscript is now acceptable for publication, you may indicate that here to bypass the “Comments to the Author” section, enter your conflict of interest statement in the “Confidential to Editor” section, and submit your "Accept" recommendation.

Reviewer #1: All comments have been addressed

2. Is the manuscript technically sound, and do the data support the conclusions?

Reviewer #1: Yes

3. Has the statistical analysis been performed appropriately and rigorously? 

Reviewer #1: Yes

4. Have the authors made all data underlying the findings in their manuscript fully available?

Reviewer #1: Yes

5. Is the manuscript presented in an intelligible fashion and written in standard English?

Reviewer #1: Yes

6. Review Comments to the Author

Reviewer #1: The authors have satisfactorily addressed my comments and have improved the clarity of the manuscript.

7. PLOS authors have the option to publish the peer review history of their article (what does this mean?). If published, this will include your full peer review and any attached files.

Reviewer #1: No

---

## [Editor Report · Acceptance letter]

4 Nov 2022

PONE-D-22-15718R1 

Untargeted metabolomics confirms the association between plasma branched chain amino acids and residual feed intake in beef heifers 

Dear Dr. Cantalapiedra-Hijar:

I'm pleased to inform you that your manuscript has been deemed suitable for publication in PLOS ONE. Congratulations! Your manuscript is now with our production department. 

Kind regards, 

on behalf of

Prof. Dr. Arda Yildirim 

Academic Editor

PLOS ONE